# Evidence of Reelin Signaling in GBM and Its Derived Cancer Stem Cells

**DOI:** 10.3390/brainsci11060745

**Published:** 2021-06-03

**Authors:** Filippo Biamonte, Gigliola Sica, Antonio Filippini, Alessio D’Alessio

**Affiliations:** 1Dipartimento di Scienze Biotecnologiche di Base, Cliniche Intensivologiche e Perioperatorie, Università Cattolica del Sacro Cuore, Fondazione Policlinico Universitario “Agostino Gemelli”, IRCCS, 00168 Roma, Italy; filippobimo@libero.it; 2Dipartimento di Scienze della Vita e Sanità Pubblica, Sezione di Istologia ed Embriologia, Università Cattolica del Sacro Cuore, Fondazione Policlinico Universitario “Agostino Gemelli”, IRCCS, 00168 Roma, Italy; gigliola.sica@unicatt.it; 3Dipartimento di Scienze Anatomiche, Istologiche, Medico Legali e dell’Apparato Locomotore, Unità di Istologia ed Embriologia Medica, Sapienza Università di Roma, 00161 Roma, Italy; antonio.filippini@uniroma1.it

**Keywords:** cancer stem cells, glioblastoma, peritumoral tissue, reelin

## Abstract

Glioblastoma (GBM) is the most aggressive and malignant form of primary brain cancer, characterized by an overall survival time ranging from 12 to 18 months. Despite the progress in the clinical treatment and the growing number of experimental data aimed at investigating the molecular bases of GBM development, the disease remains characterized by a poor prognosis. Recent studies have proposed the existence of a population of GBM cancer stem cells (CSCs) endowed with self-renewal capability and a high tumorigenic potential that are believed to be responsible for the resistance against common chemotherapy and radiotherapy treatments. Reelin is a large secreted extracellular matrix glycoprotein, which contributes to positioning, migration, and laminar organization of several central nervous system structures during brain development. Mutations of the reelin gene have been linked to disorganization of brain structures during development and behavioral anomalies. In this study, we explored the expression of reelin in GBM and its related peritumoral tissue and performed the same analysis in CSCs isolated from both GBM (GCSCs) and peritumoral tissue (PCSCs) of human patients. Our findings reveal (i) the higher expression of reelin in GBM compared to the peritumoral tissue by immunohistochemical analysis, (ii) the mRNA expression of both reelin and its adaptor molecule Dab1 in either CSC subtypes, although at a different extent; and (iii) the contribution of CSCs-derived reelin in the migration of human primary GBM cell line U87MG. Taken together, our data indicate that the expression of reelin in GBM may represent a potential contribution to the regulation of GBM cancer stem cells behavior, further stimulating the interest on the reelin pathway as a potential target for GBM treatment.

## 1. Introduction

GBM is a malignant and a very aggressive form of primary brain cancer whose clinical treatment represents a major challenge based on its great heterogeneity and the high frequency of recurrences. Despite the aggressive treatment of patients with a combination of surgery followed by radio and chemotherapy, GBM remains essentially an incurable disease with a median survival time ranging from 12 to 18 months [1,2,3]. Although most studies related to GBM are focused on the core tumor area, increasing interest has recently raised toward the involvement of the peritumoral area of GBM [4,5,6,7,8,9,10,11,12]. Infiltrating tumor cells into this apparently healthy parenchyma appears critical for the onset of recurrence, causing the surgical resection of the tumor to be fruitless. Therefore, there is an urgent need to identify novel markers both in GBM and in the peritumoral tissue that may be beneficial to counteract the progression of the disease. Reelin is a large secreted extracellular glycoprotein of approximately 400 kDa [13,14] that plays a key role in the regulation of neuronal migration during mammalian brain development [15]. Studies performed in the null reeler mouse have suggested that the serine protease activity of reelin is crucial for its function in the developing brain [16]. See [17,18] for a detailed review of the main structural and functional features of the reelin glycoprotein in the nervous system development. Signaling mechanisms triggered by reelin depend on the recruitment of distinct cell surface receptors, i.e., very low density lipoprotein receptor (VLDLR), apolipoprotein E receptor 2 (ApoER2 or LRP8) [19,20,21,22], α3β1 integrin [23,24,25], and members of the cadherin-related neuronal receptor (CNR) family [26]. ApoER2 and VLDLR show the highest affinity for reelin [27] and mediate their intracellular signaling following src-mediated tyrosine phosphorylation of the intracellular adapter protein disabled-1 (Dab1) [19,28,29,30,31,32], which in turn activates a number of downstream adaptor proteins, including Crk, C3G, Rap1, and phosphatidylinositol-3-kinase (PI3K) [25,31,33,34,35,36]. Interestingly, a number of studies has reported changes in the expression of reelin in different cancer types even outside the nervous system [37]. Reelin expression has been found reduced in breast [38,39,40], colorectal [41,42], and pancreatic cancers [43], while it has been found to be increased in retinoblastoma [44], myelomas [45], and prostate cancers [39,46,47]. Here, we investigated the expression of reelin in human tissue samples obtained from tumoral lesion and the peritumoral area of GBM. We also investigated the expression of both reelin and its adapter molecule Dab1 in CSCs isolated from GBM (GCSCs) and from the peritumoral tissues (PCSCs) derived from GBM patients as previously reported [46] and further characterized by our group [4,5,6,7,9,10,11]. Finally, to explore the potential biological role of reelin in GBM, we studied its effect on the migration of human glioma cell line U87MG. Although the precise molecular mechanisms are still unclear, our data suggest an interesting role of secreted reelin in stimulating the migratory capability of GBM CSCs, therefore opening a new avenue in the investigation of the potential biological role of reelin in GBM.

## 2. Materials and Methods

### 2.1. GBM Tissue and Cell Cultures

Isolation and characterization of neurospheres derived from both samples of GBM (glioblastoma cancer stem cells, GCSCs) and peritumoral tissue (peritumoral tissue cancer stem cells, PCSCs) were performed by Prof. A. Vescovi and his group [46,47]. The detailed methodology was also reported in our previous papers [9,10,11]. GCSCs and PCSCs were cultured in a NeuroCult™ NS-A Proliferation Kit (Stemcell Technologies Inc, Vancouver, BC, Canada) supplemented with 20 ng/mL human recombinant EGF, 10 ng/mL human recombinant bFGF, and 2 ug/mL heparin (all from StemCell Technologies Inc.). The U87MG grade IV glioma cell line was maintained in DMEM containing 10% (*v*/*v*) fetal calf serum, 200 mM L-glutamine, and 100 units/mL penicillin/streptomycin (Life Technologies Carlsbad, CA, USA). All cell types were maintained at 37 °C in a 5% CO_2_ humidified atmosphere. Recombinant reelin was purchased from R&D Systems (Minneapolis, MN, USA).

### 2.2. Immunohistochemistry and Stereological Analysis of GBM and Peritumoral Tissue

Paraffin-embedded tissue sections from tumor and peritumoral samples of four patients were used for immunohistochemical analysis. In brief, after blocking endogen peroxidase, histological sections were incubated with either monoclonal anti-reelin MAB5364 (Millipore) or with appropriate IgG1 control antibody overnight at 4 °C. Specimens were then incubated with an HRP/Fab polymer conjugate (SuperPicTure Polymer Detection Kit, Invitrogen, Camarillo, CA, USA). Immunopositive cells were visualized by brown DAB (Vector Laboratories, Inc., Burlingame, CA, USA) staining and sections were lightly counterstained with Mayer’s hematoxylin. For accurate quantification of immunopositive cells, we employed an unbiased stereological technique by means of the Stereo Investigator System (Stereo Investigator software, Version 9.14© 2010, MicroBrightField Europe, Magdeburg, Germany) as previously described [9,48,49]. This technique allows quantitative and unbiased evidence of a three-dimensional structure to be obtained, based on the analysis performed on two-dimensional histological sections [50]. The cellular density of reelin-negative (−) and reelin-positive (+) cells was determined by dividing the number of cells from three different fields by the area of region of interest (ROI) of GBM and peritumoral tissue with the Stereo Investigator software (Version 9.14© 2010, MicroBrightField Europe, Magdeburg, Germany).

### 2.3. Quantitative RT-PCR of Reelin Expression in GCSCs and PCSCs

For real-time PCR analysis of reelin expression, total RNA was extracted from GCSCs and PCSCs with the TRIZOL Reagent (Life Technologies Corporation, Gaithersburg, MD, USA) according to the manufacturer’s instructions. Total RNA was therefore retro-transcribed into single-stranded cDNA by a standard 20-µL RT reaction with the High-Capacity cDNA Reverse Transcription Kit (Applied Biosystems, Foster City, CA USA). cDNA generated from the reverse transcription reactions was amplified by real-time PCR with the SensiMix SYBR kit (Bioline, London, UK) in a total volume of 20 µL with the Step One Real-Time PCR System (Life Technologies Corporation). The following primers couples were used for PCR reactions: reelin, 5′-AAATGAGCATGGGCTGTAGCAAGC-3′ and 5′-ACTCTTCGGTGACAAGATGCCAGT-3; Dab1, 5′-TGAAACTCAAGGGCGTTGTTGCTG-3′ and 5′-TCCAAAGGCCCGGTGATCTGTAAT-3′; β-Actin, 5′TGCACCACACCTTCTACAATGA-3′ and 5′CAGCCTGGATAGCAACGTACAT-3′. The level of gene expression was expressed as the relative fold change vs. the β-actin messenger RNA using the ΔΔ_Ct_ method [51] using the Step One System Software (Life Technologies Corporation).

### 2.4. Evaluation of U87MG Cell Migration

The evaluation of cell migration was performed using the modified Boyden chamber assay with 8.0 µm polycarbonate membrane inserts (BD Biosciences, Bedford, MA, USA). A suspension of 200,000 U87MG cells was added to the upper well of the Boyden chamber and 500 µL of GCSCs- or PCSCs-derived conditioned medium were added to the lower well either in the presence or absence of 200 ng/mL of recombinant reelin. Cells were incubated under the described conditions for 18 h in a 37 °C, 5% CO_2_ incubator. At the end of the incubation, membranes were rinsed in PBS, fixed by 3.7% formaldehyde for 10 min at room temperature, and permeabilized by 100% methanol for 20 min at room temperature. After scraping off non-migrated cells, polycarbonate membranes were stained with DAPI and migrated cells were counted under a fluorescence microscope (Nikon Eclipse TS100, Nikon, Tokyo, Japan). The number of cells, counted from five different microscopic fields for each sample, was then averaged and graphed.

### 2.5. Statistical Analysis

Statistical analysis was performed by one-way ANOVA, using the Sigma Stat software package (version 3.1 for Windows). Differences between groups were tested by the Bonferroni-corrected *t*-test.

## 3. Results

### 3.1. Expression of Reelin in GBM

To investigate the expression of reelin, we first performed immunohistochemistry analysis on both GBM and peritumoral tissue specimens. This analysis clearly revealed the presence of a high number of reelin-positive cells in GBM compared to the peritumoral tissue (Figure 1A,B). Moreover, reelin was also found to be expressed at the level of endothelium in both GBM and peritumoral tissue, in agreement with previous findings of reelin expression in brain endothelial cells [48]. Due to the structural complexity of samples studied and to ensure rigorous and unbiased quantitative estimation of data generated by immunohistochemistry, we turned to the stereological analysis [9]. This analysis demonstrated that the cell density (cells/mm^2^) of reelin (+) cells was significantly (*p* = 0.015) higher in GBM than in peritumoral tissue (Figure 1C). Our data are consistent with the expression of reelin in both GBM and the corresponding peritumoral area, and with a higher cell density revealed in GBM. These data are in apparent contrast with data published by others that reported a diminished expression of reelin in GBM samples compared to non-pathological samples. We believe that this discrepancy may be due to the extreme heterogeneity of samples employed in the above-mentioned work [49]. Indeed, while our analysis was performed only on grade IV GBM samples, the previous study included primary and secondary glioblastomas and different types of astrocytoma.

### 3.2. Expression of Reelin and DAB1 mRNA in GCSCs and PCSCs

We next investigated, by real time-PCR (qPCR) analysis, the expression of both reelin and its adaptor molecule Dab1 mRNA in both GCSCs and PCSCs isolated from four different GBM patients. Our results demonstrated a marked expression of both reelin and Dab1 transcripts in both GCSCs and PCSCs. However, while reelin mRNA expression was clearly higher in PCSCs compared to GCSCs in all the samples analyzed, the expression of Dab1 mRNA appeared more heterogeneous, and was upregulated in three out of four PCSC samples (Figure 2). This data indicated that both GCSCs and PCSCs are endowed with the ability of expressing reelin, although at a different extent. Moreover, the expression of Dab1 mRNA suggests the possible activity of the reelin pathway in PCSCs rather than GCSCs.

### 3.3. Effects of Reelin and CSCs Culture Medium in the Migration of Human Malignant Glioma Cell Line U87MG

To deepen the significance of reelin expression in CSCs, we then investigated the potential effect of reelin released by GCSCs and PCSCs. Therefore, we used both GCSC- and PCSC-derived conditioned media (CM) to treat a target tumor cell type, i.e., U87MG in a Boyden chamber assay. The results of these experiments showed that PCSC- and PCSC-derived CM were able to induce migration of U87MG only in the presence of 200 ng/mL recombinant reelin (Figure 3A–F). These data indicated a possible disorganized secretory mechanism in CSCs that prevented the release of soluble reelin into the culture medium. Notably, in the presence of recombinant reelin, we also observed the formation of small spheroid-like structures (Figure 3B,D,F), similar to what was already reported in neurons and glial cells [52], indicating that U87MG cells were responsive to reelin stimulation. Data obtained from different assays were collected, analyzed, and graphed (Figure 3G).

## 4. Discussion

The glycoprotein reelin has been demonstrated to be critical during the positioning of migrating post-mitotic neurons and the laminar organization of several central nervous system (CNS) structures during development [13]. Moreover, it has been found transiently expressed both in the peripheral nervous system (PNS) [53] and other tissues [54,55,56]. The idea that reelin may function outside of the CNS and PNS has been supported by recent works that have linked the reelin signaling pathway to different cancers, such as prostate, gastric, and breast cancers. However, clear evidence of the involvement of reelin in tumor development has not been fully elucidated. Here, we provide evidence that reelin is expressed in both GBM and its adjacent peritumoral tissue at a different extent. Interestingly, our immunohistochemical analyses on GBM samples and peritumoral tissue indicate the presence of reelin in both areas of the brain and suggest a possible activity of the reelin pathway in GBM progression. The higher expression of reelin in GBM may indicate a possible contribution of the protein in the regulation of cell survival. It has been, indeed, demonstrated that reelin functions as an anti-apoptotic factor, promoting cell survival through the Dab1-activated Akt pathway. In addition, PI3-kinase inhibitors counteract reelin-induced cell survival [57]. Notably, although CSCs have been detected in many types of gliomas, few studies have investigated their significance in GBM adjacent peritumoral tissue [9,10,11]. Our data on the expression of the major adaptor molecule of reelin, i.e., Dab1, highlight the involvement of reelin pathway in the GCSCs and PCSCs of GBM. However, the precise mechanism remains elusive. In this respect, we report here the expression of both reelin and Dab1 mRNAs in GCSCs and PCSCs. Specifically, reelin transcript was found to be expressed higher in PCSCs compared to GCSCs, whereas the expression of Dab1 mRNA appeared more heterogeneous within pairs, being overexpressed in three out of four PCSCs samples. These findings are in accordance with the high heterogeneity of CSCs couples derived from GBM patients, as previous reported [9,10]. In addition, the higher expression of both reelin and Dab-1 transcripts in PCSCs compared to GCSCs may suggest a crucial function of reelin as a potential prognostic marker for a new therapy of GBM. On the other hand, the expression of a detectable level of reelin in tissues along with the expression of both reelin and Dab1 mRNA in GCSCs, particularly in PCSCs, prompted us to investigate the biological significance of reelin signaling in GBM. Our in vitro experiments show that administration of 200 ng/mL recombinant reelin to U87MG for 18 h in culture, triggered cell migration and stimulated the formation of detectable spheroids, supporting previous findings [57]. Interestingly, when cultivated with conditioned medium collected from both GCSCs and PCSCs in culture without recombinant reelin, we did not observe marked U87MG cell migration. These results raise questions regarding the possible anomaly of the secretory mechanism of reelin or the occurrence of an aberrant post-translational modification of the protein that prevents it from being correctly released from CSCs. Taken together, our results indicate that: (a) the expression of reelin protein is remarkably higher in GBM tissue and PCSCs; (b) the expression of reelin transcript is higher in PCSCs rather than GCSCs, while the Dab1 mRNA level is more heterogeneous, being highly expressed in three out of four PCSCs samples; and (c) administration of reelin to U87MG induces migration as well as spheroid-like structure formation in vitro. Further investigation is necessary to understand the contribution of reelin-induced signaling in the maintenance of PCSC and GCSC stem cell behavior. Therefore, we hypothesize a possible function of reelin in the regulation of the invasiveness of CSCs, specifically from the peritumoral tissue, which can be prospective for the development of new therapeutic approaches for the treatment of GBM patients.

## 5. Conclusions

The extracellular matrix glycoprotein reelin plays a crucial role during mammalian brain development by contributing to the regulation of neuronal migration, establishment and branching of dendrites, synaptogenesis and synaptic plasticity. Reelin has been as well associated to some human brain disorders in the adulthood, such as lissencephaly, autism, mental disorders and Alzheimer’s disease. However, the involvement of reelin signaling in brain tumors has been poorly investigated. In this study we observed a higher expression of reelin and Dab1 transcripts in both peritumoral area and peritumoral-derived CSCs, with respect to the tumor core. Taken together, our findings might suggest a potential involvement of the reelin signaling in GBM pathology and possibly in the onset of tumor recurrence that frequently originate from the peritumoral region. Therefore, we believe that deepening the role of reelin signaling in GBM may be crucial to develop novel treatment opportunities.

## Figures and Tables

**Figure 1 brainsci-11-00745-f001:**
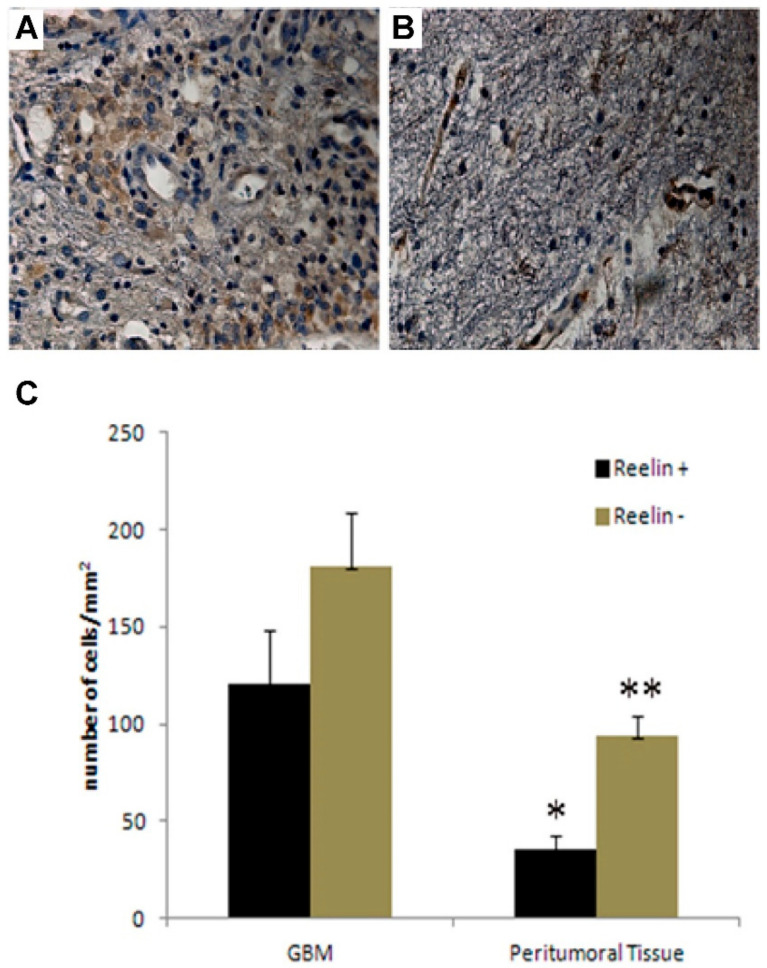
Expression of reelin and stereological analysis of GBM and peritumor tissue. (**A**,**B**), Immunohistochemical analysis of reelin expression in GBM (**A**) and peritumor tissue (**B**), representative of 4 patients. In GBM, reelin immunopositivity was detected in the cytoplasm of tumor cells and in endothelial cells. (**B**), reelin immunolocalization in the endothelium, and in the cytoplasm of apparently normal cells of peritumor tissue. (**C**) Stereological analysis of cell density in both GBM and peritumor tissue from 4 different patients. Cell density was determined by dividing cell counts by the area of the ROI, (* *p* = 0.02; ** *p* = 0.015). Bars show the mean of cell counts ± standard deviation of 4 GBM pairs.

**Figure 2 brainsci-11-00745-f002:**
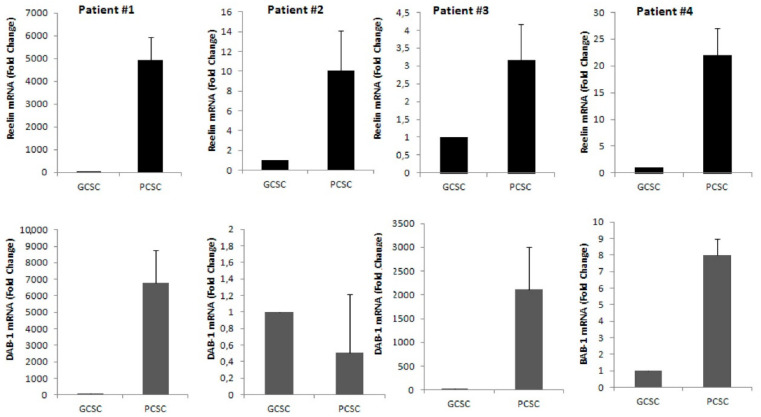
Expression of reelin and Dab1 mRNA by qRT-PCR in GCSCs and PCSCs of four patients. The level of mRNAs of reelin and Dab1 were evaluated by qRT-PCR as described in the Section 2. Upper panels show that mRNA of reelin was higher in all PCSCs compared to GCSCs. Lower panels show that mRNA expression of Dab1 was found higher in three out four examined samples of PCSCs. Bars show the mean of mRNA expression data ± standard deviation from three experiments.

**Figure 3 brainsci-11-00745-f003:**
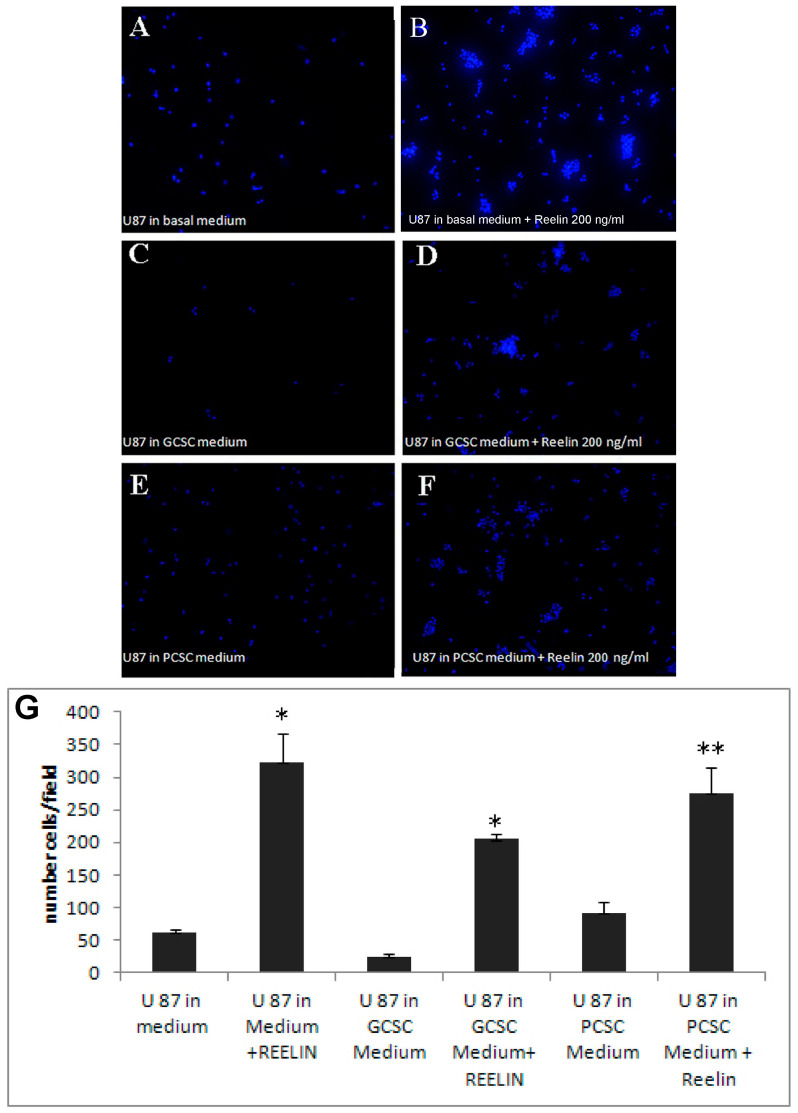
Effects of reelin and CSCs culture medium on U87MG migration. Cells were allowed to migrate for 18 h across the membrane in the Boyden chamber assay in the presence or absence of recombinant reelin. * *p* < 0.001 ** *p* < 0.05. (**A**–**F**), representative immunofluorescence images of migrated and DAPI-stained U87MG. (**G**), shows the average of migrated cells incubated with conditioned medium with or without 200 ng/mL recombinant reelin. Results are expressed as the number of cells/field; * *p* < 0.001 ** *p* < 0.05. Bars show the mean of migrated cells ± standard deviation from three experiments.

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
