# Peer review of "Evidence of Reelin Signaling in GBM and Its Derived Cancer Stem Cells"

_brainsci, 2021, doi:10.3390/brainsci11060745_

Round 1

Reviewer 1 Report

The authors have attempted to study the expression and role of reelin in glioblastoma and its peritumoral region using glioblastoma patient tissues and the patient-derived U87MG model. I strongly recommend this article for rejection due to the following reasons:

Major concerns:  1) The conclusions contradict the results: For example: In Figure 2 it is shown that Reelin mRNA expression is higher in peritumoral cancer stem cells (PCSCs), however in the discussion the following is mentioned. "Specifically, reelin transcript was found higher expressed in glioblastoma cancer stem cells (GCSCs), " 

2) The authors contradict their own conclusions: For example: in the abstract " Our study revealed i) the expression of reelin in both GBM and its peritumoral tissue by immunohistochemical analysis". However in the discussion, exactly the opposite is mentioned: "Our results indicate for the first time that : a) reelin is highly expressed in GBM but not in peritumoral tissue"

The first two points above are obvious concerns; it shows that the authors are not sure of what they are discussing. 

3) The manuscript is based on GCSC and PCSC, however the procedure for GCSC and PCSC isolation is given elsewhere. 

4) mRNA data for all 4 patients is shown , however for IHC one representative patient is shown. Why? 

5) The authors should have performed the migration assay in increasing doses of reelin. 

6) The authors state “Finally, to explore the potential biological role of reelin in GBM, we studied the effect of reelin-derived GCSCs and PCSCs”. What are reelin-derived GCSCs and PCSCs? 

7) The authors should mention about the basal levels of reelin in U87MG.  

Author Response

1) The conclusions contradict the results: For example: In Figure 2 it is shown that Reelin mRNA expression is higher in peritumoral cancer stem cells (PCSCs), however in the discussion the following is mentioned. "Specifically, reelin transcript was found higher expressed in glioblastoma cancer stem cells (GCSCs), " 

We deeply apologize with the reviewer for this evident slip-up carelessness. We have accurately revised and rephrased both the abstract and the discussion according to the reviewer’s comments. The changes made are highlighted in the original manuscript.

2) The authors contradict their own conclusions: For example: in the abstract " Our study revealed i) the expression of reelin in both GBM and its peritumoral tissue by immunohistochemical analysis". However, in the discussion, exactly the opposite is mentioned: "Our results indicate for the first time that: a) reelin is highly expressed in GBM but not in peritumoral tissue."

We would like to express our deep apologize again with the reviewer for our carelessness. We have accordingly rephrased all the parts of the ms. as indicated by the reviewer. The changes made are highlighted in the original manuscript.

3) The manuscript is based on GCSC and PCSC, however the procedure for GCSC and PCSC isolation is given elsewhere. 

PCSCs and GCSCs used in this study were previously isolated from different hospitalized patients by Vescovi's group (see ref. N 48 of the references' list) with whom we collaborated and shared the cells (see refs 9, 11, 48 and 49). Therefore, we believed it was redundant to report again the isolation procedure in the manuscript. However, if the reviewer considers it appropriate, we can include the procedure already described in the cited references into the current article.

4) mRNA data for all 4 patients is shown, however for IHC one representative patient is shown. Why? 

We believe that the stereological analysis shown in Figure 1C would have been sufficient to illustrate the overall reelin expression in formalin-fixed paraffin-embedded GBM as well as peritumor tissues. However, we thought that including a representative image of IHC, would have been of help for the readability of the results. Since the focus of our study was largely aimed at investigating CSCs (GCSCs and PCSCs), we decided to show all the PCR analysis.

5) The authors should have performed the migration assay in increasing doses of reelin. 

The dose of reelin used in this study was based on the literature. Specifically, reelin is usually employed in culture in the range of 2-100 ng/ml in various cell types (Cerebral Cortex, Volume 17, Issue 2, February 2007, Pages 294–303; Journal of Neurochemistry, 2007,103, 820–830 as examples). Since we were partially diluting the molecules into the conditional medium, we decided to use reelin at 200 ng/ml, a dose that is clearly inducing a response in U87MG cells.

6) The authors state “Finally, to explore the potential biological role of reelin in GBM, we studied the effect of reelin-derived GCSCs and PCSCs”. What are reelin-derived GCSCs and PCSCs? 

We thank the reviewer for her/his comment. We would like to better explain these experiments. Based on the expression of reelin on CSCs (significantly higher in PCSCs) we wanted to test the possible biological function of reelin on a suitable cellular target, i.e., the U87MG cell line. Our results show that, although the expression of the transcript in CSCs, these cells were unable to secrete sufficient amount of reelin protein into PCSCs (and GCSCs) conditioned medium. We can hypothesize that this is due to the malfunctioning of the releasing mechanisms, to rapid degradation of the protein or due to abnormal post-translation modification of the protein itself. As a secreted extracellular protein, reelin contains a signal peptide that functions to drive its translocation to the cellular membrane for its release. We cannot exclude that this process is less efficient or prevented in the tumor cells we investigated. We briefly discussed this hypothesis in the revised Discussion section. However, the addition of the recombinant reelin to the conditioned medium confirms that the target cells can respond to the protein specifically activating a migrating.

7) The authors should mention about the basal levels of reelin in U87MG.  

We thank the reviewer again for her/his comment. The choice of U87 to be employed as target cells in our experiments was based on the study by Schulze and collaborators (see reference n. 55, already included in the first version of the manuscript). To this regard, in figure 3C, D, E and F, authors demonstrate that neither reelin nor Dab1 messenger RNAs were expressed in U87. By contrary, they do express both VLDLR and (at higher level) APOER2 receptor, indicating that U87 can respond to reelin. Based on these findings, we decided to use U87 in our experiments.

Reviewer 2 Report

The authors Biamonte et. al., have studied the role of Reelin in the signaling pathway of GBM and have demonstrated its role in the cell migration of U87MG cells. 

The research is relevant to the advancements in the field and adds value to the current understanding of GBM.

However, I am curious to know if the authors had performed any western blot analysis to identify the most possible downstream pathway for cancer propagation.

Author Response

However, I am curious to know if the authors had performed any western blot analysis to identify the most possible downstream pathway for cancer propagation.

We truly thank the reviewer for her/his appreciation of our study. Concerning the possible downstream pathways activated by reelin in CSCs isolated from GBM, due to the high (and reported) heterogeneity of GBM samples as well as to the complexity of signaling pathways activated by reelin, we are not able at this time to draw a conclusive scenario of the downstream pathways in our experimental model. We are currently exploring these mechanisms by performing in vitro analysis aimed at specifically targeting reelin and its adaptor molecule Dab1 in both GCSCs and PCSCs.

Round 2

Reviewer 1 Report

Thank you for the revised manuscript and cover letter.

3) If there is space I would suggest to include the PCSC and GCSC isolation procedure in the current article. 

6)"What are reelin-derived GCSCs and PCSCs?".

Thank you for the explanation for this question. I meant that it should be 'reelin-derived from GCSCs and PCSCs' or 'CSCs-derived reelin'. This has now been corrected in the updated manuscript. 

Author Response

We would like to thank the reviewer for his/her appreciation of the revised manuscript. We would like to point out the following. Since the methodology of isolation of CSCs from both GBM and peritumoral surgical samples was conceived and performed by Prof. Vescovi (and collaborators), which was not involved in the current study, we preferred to report the original references in the current manuscript. In other words, we wanted to avoid any misunderstanding concerning the original procedure's authorship.  However, to better clarify this point, we have further modified the related section of the Materials and Methods, as it has been highlighted in the text (see page 2). We hope to have satisfactorily answered the reviewer's comment.